# For Children or Grandchildren?—The Motivation of Intergenerational Care for the Elderly in China

**DOI:** 10.3390/ijerph20021441

**Published:** 2023-01-12

**Authors:** Juan Luo, Mengmeng Cui

**Affiliations:** School of Management, Shanghai University of Engineering Science, Shanghai 201620, China

**Keywords:** alternate generation care, economic support, spiritual comfort, exchange motivation, altruism motivation

## Abstract

Considering our aging society and declining birthrate, we studied the motivations for intergenerational care for the elderly in China and analyze it through different generations of children and grandchildren. This paper mainly uses logistic regression analysis, descriptive statistics, heterogeneity analysis, robustness analysis, and other quantitative methods to analyze the data of the China Household Tracking Survey (CFPS) in 2018. According to the relevant research results, we believe that both children or grandchildren may be motivations. However, the proportion of factors is higher because the proportion of parents choosing alternate-generation care has increased based on the financial support provided by their children, and underdeveloped children are more likely to be favored by their parents to provide alternate-generation care. This paper is more inclined to consider children to be the main motivator for intergenerational care under the dual motives of “feedback” exchange and altruism through “helping the weak”. Next, the influence factors of a series of control variables are analyzed for the groups that take care of the next generation, and it is found that the intimacy between the elderly and their spouses, the number of meals with their families per week, and the use of electronic products will all affect the degree of care provided by their fathers. Based on this, this paper proposes that the three forces of government, society, and family are needed for joint support of the elderly when they are taking care of each other. We will make generational care less responsible, free and burdensome, and at the same time respect the right of the elderly to say “no” to providing care.

## 1. Introduction

At the end of 2021, the proportion of China’s population aged 60 and above was 267.36 million, accounting for 14.2% of the national population, which indicates that China has an aging society. However, the contrasting statistic of only 10.62 million births, and a natural population growth rate of 0.34%, show that the characteristics of a “declining birthrate” are becoming increasingly obvious. Against this background, the elderly in China play an important role in caring for their grandchildren. Intergenerational care is a prominent embodiment of the immediate and reliable function of “caring for the young and helping the young” in family security, which includes three generations of subjects; namely, parents, children, and grandchildren, and their multi-level intergenerational relationships.

Wang, H.Y (2018) and Fei, X.T (1986) reported that intergenerational care, as a contemporary widespread and profound social phenomenon, has broken the reciprocal balance of the feedback mode of both nurturing and support in the traditional intergenerational relationship between parents and children in China [1,2]. This leads us to consider the possibility that, based on the intergenerational care of the elderly, the financial support and care provided by children to their parents is more motivated by an exchange, while the care provided by the elderly to their grandchildren is not only motivated by the exchange of intergenerational support provided by their children, but also motivated by blood relationship and emotional altruism to a greater extent. Therefore, by analyzing the influencing factors of intergenerational care for the elderly in different generations of children and grandchildren, it was found that the intimacy between the elderly and their spouses, and the recognition of their children’s promise, will all affect the degree of intergenerational care provided by their parents.

## 2. Data and Methods

### 2.1. Literature Review and Research Hypothesis

A large number of studies have shown that there are many factors influencing the intergenerational care of the elderly. Yan, Y. et al., (2022) found that, compared with the elderly who did not provide intergenerational care, the elderly who provided intensive care were more likely to participate in society [3]. Chen, C. et al., (2022) believed that intergenerational care for grandchildren could improve the well-being of the elderly, but a strong intensity of intergenerational care would reduce it [4]. Wang, L.J. (2022) found that there was a negative correlation between the intensity of care and mental health. With the increase in the intensity of care, the risk of depression in middle-aged and elderly people gradually increased [5]. Buchanan, A. and Rotkirch, A. (2018) et al., believed that intergenerational care can be called a “scarce” resource in contemporary Chinese society. Intergenerational caregivers are the real “unsung heroes” in supporting their children’s labor participation and even in deciding whether to have a second child [6].

From the perspective of generations, the lower the income, health level, and education level of offspring, the easier it is to receive the support of parental intergenerational care. At the same time, the financial support provided by offspring will increase the amount of time devoted to parental intergenerational care, as studied by Jin, W.L. (1986) [7]. Intergenerational care embodies the exchange of economic support and care between generations. At the same time, it enhances the intimacy with grandchildren so as to further enhance the feedback of children’s old-age support, it also promotes intergenerational care for parents, as studied by Zhong, Z.B. et al., (2015) [8]. During the process of grandchildren’s birth to their upbringing, the intergenerational focus of family structure also naturally shifts from the father supported by the children to the grandchildren, becoming a new force for family development, as studied by Yan, Y. [9]. With the development of modernization, although the family structure tends to be smaller, the proportion of interaction between generations is on the rise. Raising grandchildren has become a business that needs to be completed by two generations of fathers and sons, as studied by Li, T. and H, W.B. (2018) [10]. The provision of intergenerational care for the elderly is a continuous manifestation of self-worth, which helps them to maintain more social contact and exercise behaviors, as studied by He, N. (2021) [11].

From the perspective of urban and rural areas, it was found that, compared with their relatives living in rural areas, parents in urban areas will provide more intergenerational care. This may be because the rural elderly are still busy with farm work and lack enough time and energy to take care of their grandchildren, while the urban elderly have the double guarantee of pension and leisure time, as studied by Jiang, W.G. and Liu, W.H. (2021) [12]. Although the rural elderly care for their grandchildren, they are happy, but they will also feel physical and mental pressure and require the active guidance and full support of the government, society, and their family, as studied by Sun, Y.T et al. [13]. After the rural elderly receive a pension, they can change the amount of financial support provided by their offspring, reducing the possibility of living with their offspring, and thus reducing the enthusiasm of providing care for the next generation, as studied by He, Y.L. (2021) [14]. Some scholars have found that intergenerational care is an intermediary variable in rural areas, and the elderly’s participation in social insurance will indirectly affect their own financial support through intergenerational care, as studied by Zheng, J.H (2021) [15].

From the perspective of gender, women are often the bearers of family burdens in intergenerational support and intergenerational care. Relevant government departments and social organizations should introduce relevant policies to make them share the fruits of social and economic development more fairly, thus reducing the double pressure on women from society and their families, as studied by Chen, J. (2020) [16]. In addition, compared with the female elderly who provide non-moderate care, the female elderly who provide moderate care have a relatively high level of mental health, as studied by Wu, J. (2021) [17]. Due to the influence of China’s traditional family culture, relatives will be influenced by social cultural background and social norms when choosing whether to provide intergenerational care. For example, in the culture of filial piety, Song, L. and Feng, X. (2018) found that the responsibility of supporting the elderly in most families is borne by the sons, and that daughters are more involved in supporting their in-laws after marriage. At the same time, in this state, compared with the daughter’s children, the son’s children deserve more care and assistance from grandparents [18]. Additionally, the fertility desire of have two children will also affect the enthusiasm of the elderly to provide intergenerational care. Parents cannot provide intergenerational care, which makes it difficult for offspring to take care of work and children, thus reducing the willingness of women of childbearing age in the workplace to bear two children, as studied by Zhang, G.H. and Shi, Y.Y. (2021) [19].

In summary, a large number of scholars have studied and analyzed the influence of intergenerational relations, the differences between urban and rural areas, gender, and other aspects of intergenerational care. At the same time, some scholars have proposed that the intergenerational support provided by offspring and the emotional support obtained from caring for grandchildren will promote the judgment of intergenerational care for the elderly. However, while some scholars have studied both offspring and grandchildren in these studies, none have classified and compared the motivation of intergenerational care for the elderly in China. Based on this, we put forward hypothesis 1: the “son” level, such as the financial support provided by offspring to their parents and the closer relationship between the children and their parents, will encourage parents to provide care for the next generation, because the “son” is a “helping the weak” behavior based on a relationship of exchange and blood relationship. Hypothesis 2, meanwhile, is that the “grandchild” level—for example, the elder’s providing care for the next generation—will be affected by the gender, age, and happiness of the grandchild, because the “grandchild” is a “caring for the young” behavior based on blood relationship.

### 2.2. Data Sources

The China Family Follow-up Survey (CFPS) data were implemented by the China Social Science Research Center of Peking University. Since 2010, more than 16,000 families in 25 provinces in China have been followed up. We selected the 2018 data from this survey. The survey content involves detailed information, such as income level, education level, age, marital status, occupation type, family relationship, and care of children at the family and individual levels. After removing some missing variables and irrelevant variables, this paper combined children data with personal data and finally selected 1267 family samples.

### 2.3. Model Setting and Empirical Methods

This study adopts the perspective of intra-family comparison to establish an analytical framework of intergenerational care motivation (Figure 1). Intergenerational care for the elderly is influenced by two levels of factors: offspring and grandchildren, and other related control variables concerning parents, offspring, and grandchildren. At the offspring level, variables include financial support provided by offspring, care support provided by offspring, and intimacy between parents and offspring, etc. At the level of grandchildren, factors include grandchildren’s age, grandchildren’s gender, and the elderly’s sense of happiness, etc. In this study, the fixed-effect model is adopted to control the influence of unobserved heterogeneity at the offspring and grandchildren levels to the maximum extent. The specific expression is as follows:
(1)$${\mathrm{take}}_{\mathrm{care}}=\beta 1x1\mathrm{hi}+\beta 2x2\mathrm{hi}+\beta 3x3\mathrm{hi}+\beta 4x4\mathrm{hi}+\beta 5x5\mathrm{hi}+\beta 6x6\mathrm{hi}+\alpha h+\varepsilon$$

Among these variables, take_care_ is the dependent variable of day care of this study, which means that the parents of family H provide alternate-generation care for their ith child. In turn, 1 is the estimated coefficient of variable *x*1 (financial support provided by offspring), *β*2 is the estimated coefficient of variable *x*2 (caring support provided by offspring), *β*3 is the estimated coefficient of variable *x*3 (intimacy with children), *β*4 is the estimated coefficient of variable *x*4 (age of grandchildren), *β*5 is the estimated coefficient of variable *x*5 (age of grandchildren), and *β*6 is the estimated coefficient of variable *x*6 (happiness of the elderly).

In this study, the logistic regression method was used for descriptive statistics, multidimensional analysis, heterogeneity analysis, and robustness analysis, constructing a technology roadmap as shown in Figure 2.

## 3. Empirical Results

### 3.1. Descriptive Statistics

In this paper, daycare was selected as the dependent variable, explanatory variables were divided into "for children" and "for grandchildren", and "promising children", "whether to use mobile phones" and "education level of the elderly" were selected as the control variables. See Table 1.

#### 3.1.1. Description of Dependent Variables

According to the selected data, it was found that intergenerational care accounted for a certain proportion of all types of care, among which the proportion of day care for the elderly was 42.4%, while the proportion of evening care was 37.4%. Among this, the proportion taking care of children in primary school in the daytime was 61.2%, followed by preschool children at 29.0% and middle school at 9.9%. In the evening, the proportion taking care of children in primary school was 63.3%, followed by preschool children at 26.1% and middle school at 10.6%. This shows that the “child-care function” of intergenerational care is significant for grandchildren. In addition, the proportion of intergenerational care in daytime and evening is close to the level, and there is no obvious difference. See Table 2.

#### 3.1.2. Description of Independent Variables

At different ages of offspring, financial support generally shows an inverted U-shaped distribution, which rises first and then falls, while care support generally shows a positive U-shaped distribution, which is high at two ends and low in the middle. This is because children’s economic ability is not enough to provide sufficient financial support to their parents at an early age, but with an increase in age, the steady advancement of family and career gives them the ability to provide financial support to their parents. When these children have their own children, they will gradually invest in the next generation, and the financial support to their parents will decrease. However, at an early age, children have not yet married and spend more time caring for their parents. With an increase in age, these children gradually have their time occupied by their own families, and the time spent caring for their parents is reduced. As the next generation of children gradually becomes independent, the parents of the children also become older, and the time that the children provide parental care increases again.

### 3.2. Multi-Dimensional Analysis of Intergenerational Care, Offspring and Grandchildren

The dependent variable of the data analyzed was alternate generation care during the day, which belongs to the binary variable (0 = no day care, 1 = day care). From the “child” point of view, model 1 was set up for regression analysis, and the explanatory variables were financial support provided by offspring, care support provided by offspring, and intimacy between parents and children. It was found that financial support provided by offspring will positively promote the elderly to provide intergenerational care, while care support provided by offspring will discourage the elderly from providing intergenerational care. This shows that, at the material level, financial compensation provided by offspring to their parents will reduce the economic cost required by the elderly to provide intergenerational care. From another point of view, older people who require care support from their offspring may lack good health, so they are often unable to provide care for their grandchildren. At the same time, we further found that the closeness of the relationship between children and their parents did not affect the proportion of care provided by the elderly. However, children’s financial support for their parents increased the proportion of care provided by older people from 46.3 to 53.7%. However, when parents choose to care for their grandchildren, it is difficult for them.

From the perspective of “grandchildren”, model 2 was set up for regression analysis. The core independent variables were the gender and age of grandchildren and the sense of happiness of the elderly. The results in Table 3 show that “there is no significant correlation between intergenerational care and the gender of grandchildren” and “there is a significant correlation between intergenerational care and the age of grandchildren”. This shows that the gender of grandchildren does not determine whether the elderly provide intergenerational care or not, and to a certain extent, it shows that with societal progress, the concept of preference for boys among the elderly gradually fades. At the same time, when grandchildren are relatively young, they will receive more intergenerational care from their grandparents, which may be because the children are still in the early stage of their career and bear a series of economic costs of feeding and raising children, so they are not enough time and energy to accompany their children, and the children are young and do not have the ability to live independently. At this time, qualified grandparents will take up the responsibility of nurturing and taking more care of their children. However, there was no significant correlation between the happiness of the elderly and their intergenerational care, although the negative coefficient shows that there is the possibility that the intergenerational care of “grandchildren” will weaken the happiness of the elderly, due to shortening the time for the elderly to enjoy their own old-age life, reducing their time for participating in community activities and traveling together, etc.

Regression analysis was conducted by setting model 3 from the dual perspectives of “children” and “grandchildren”, and it was found that the results of explanatory variables and controlled variables were consistent with those of model 1 and model 2. Therefore, further analysis of the controlled variables showed that the more promising children are, the lower the degree of intergenerational care for the elderly will be, and the more meals they have with their families every week will promote the provision of intergenerational care for the elderly. At the same time, the use of smart products, such as mobile phones, by the elderly is also significant. This shows that when the elderly choose intergenerational care, they tend to “help the weak” by assisting their children, who may be in a weak position or unable to take care of their own children because of their work, and having meals with their families every week will increase the time spent with their children and grandchildren. At the same time, the use of mobile phones by the elderly will help their social communication and cost communication, thus increasing the frequency of providing intergenerational care.

Based on the above conclusions, it can be preliminarily verified that providing care for children is a behavior based on “helping the weak” through exchange and consanguineous relationships, and providing care for grandchildren is a behavior hypothesis of “caring for the young” based on a consanguineous relationship, under the conditions that security mechanisms are stable and the elderly are more willing to spend time and energy on self-improvement and social communication; however, this is restricted by the fact that their offspring are weak and their grandchildren are young, and at the same time, their offspring also provide financial support to the elderly with backward compensation and feedback. See Table 3.

### 3.3. Heterogeneity Analysis

#### 3.3.1. Heterogeneity Comparison of Parents with Different Genders

The parental results in the previous article are based on mixed samples of men and women. The samples were divided into men and women for grouping regression to further test the conclusions in the previous article. See Table 4.

Whether grandparents were male or female, the care support provided by their children and the age of their grandchildren all affected whether they took care of the next generation, which is consistent with the conclusion in the regression of the whole sample. However, in the sample of elderly men, the financial support provided by children still had a positive impact on whether the parents took care of the next generation. However, the children’s promise and the number of meals a week with the family were not significant in relation to providing care for the elderly. Meanwhile, the higher the intimacy between elderly men and their spouses, the more the elderly men will provide day care. This shows that if elderly men and their spouses have a harmonious relationship, they will take on more family affairs together. Under the traditional family structure in China, elderly women transition from raising their children to raising the next generation, bearing more burdens within the family. Some elderly women still have to shoulder the task of caring for their grandchildren even if their physical condition is inadequate, while elderly men with harmonious relationships with their spouses will be more understanding and considerate of each other and take care of each other from generation to generation.

In the female sample, the financial support provided by their children, their children’s promise, and the number of meals a week with the family were no longer significant, which shows that although some elderly women are the main force of intergenerational care, and that they lack a certain economic foundation and access to economic sources compared with older men, for older women, regardless of whether their offspring provide certain financial support or not, intergenerational care for their grandchildren is more altruistically motivated and based on kinship.

#### 3.3.2. Heterogeneity Comparison of Parental Correspondence between Urban and Rural Areas

The results of parental generation mentioned above are based on mixed samples of urban and rural areas, but there were some differences between urban and rural families in China in terms of family intergenerational structure and intergenerational care. Therefore, parents were further divided into urban and rural samples for grouping regression, and relevant results were obtained. See Table 5.

No matter the grandparents with rural household registration or non-rural household registration, the economic support and care support provided by their children and the age of their grandchildren significantly affect whether the grandparents take care of their grandchildren, and younger grandchildren are more likely to get care from their grandparents. At the same time, regardless of area, the gender of the grandchildren did not affect whether the parents took care of each other. Additionally, based on the care provided by the grandparents, the proportion of the grandsons taken care of was 50.4%, and 49.6% took care of their granddaughters. The difference is not significant, which is exactly consistent with the conclusion in the full sample regression.

In urban samples, the elderly’s sense of self-achievement and intimacy with their spouses significantly affected their daytime care, which indicates that the urban elderly who take care of each other from generation to generation will enhance their sense of self-achievement, find and identify their self-worth, and have a better psychological state. The 18th National Congress of the Communist Party of China, determined that, by the end of 2020, in terms of social security, we should establish and improve the social security systems of urban and rural areas as a whole, gradually improve the basic old-age insurance benefits for urban and rural residents, and pay attention to the protection of the rights and interests of elderly women, so that equality between men and women can be further reflected, the differences between urban and rural areas can be reduced, and the concepts of “raising children to guard against old age” and “being a good grandson” for the elderly are gradually weakened and abandoned. Rural elderly people are increasingly paying attention to their time planning and social contacts.

#### 3.3.3. Robustness Analysis

To test the robustness of the research conclusion, overnight alternate care was used instead of daytime alternate care, and the influence of explanatory variables and control variables on grandparents’ night-time alternate care was studied. The logistic model with a fixed effect was used to regress, and the results are shown in Table 6. Except for the elderly’s sense of self-accomplishment, their children’s success and the number of meals with their families no longer being significant, other key variables showed almost no difference from the main model in terms of significance, positive and negative directions, which shows that the results of this study are stable. See Table 6.

## 4. Research Conclusions

Based on the analysis of the data of the China Family Follow-up Survey (CFPS) in 2018, we found that the motivations for intergenerational care were based on the financial support provided by the offspring to the elderly, the children’s promise (“feedback” exchange motivation and “helping the weak” altruistic motivation) and the age of the grandchildren (“helping the young” altruistic motivation). However, intimacy with offspring, happiness, and the gender of the grandchildren did not affect the intergenerational care of the elderly, because according to the relevant research results, this paper demonstrates that both motivations have advantages, but they are more inclined to the former, in which “for children” is an exchange motivation based on offspring feedback and the altruistic motivation of parents to help the weak, and “for grandchildren” is an altruistic motivation based on “caring for the young”. In addition, heterogeneity analysis found that in the male sample, intimacy with a spouse had a positive impact on parental care. At the same time, regardless of urban and rural areas, the gender of grandchildren did not affect whether parents took care of each other or not, while the urban elderly had a better sense of self-achievement when taking care of each other. By matching the data and replacing the explained variables with night-time care, the results obtained were consistent with day care, which proves the robustness of the analysis results. Therefore, based on the above research conclusions, this paper puts forward suggestions at three levels: government, society, and family.

### 4.1. Government Level

#### 4.1.1. Construct the Top-Level Design of “Healthy Aging” and “Active Aging”, Focusing on the Needs of Elderly Women

The government needs to further co-ordinate innovative institutional arrangements and policy measures to adjust the labor and employment markets, particularly to build a friendly social environment for elderly women to participate in various social and economic activities. In all aspects and fields, such as policy design, project operation, and system guarantee, it should encourage elderly women to participate in social and economic activities more effectively, share the fruits of social and economic development more fairly, enhance women’s social and economic status, strive to meet the diverse needs of elderly women in medical care and old-age care, and reduce the burden of intergenerational care for elderly women in all aspects.

#### 4.1.2. Strengthen the Supervision of the Nursery Service Market and Eliminate Children’s Concerns

At present, although the traditional method of raising children in China is mainly to raise children from generation to generation when the grandparents cannot provide intergenerational care due to physical health and other reasons, they must seek help from other external forces, such as domestic nannies, without affecting women’s work. Therefore, the government should strengthen the supervision of the nursery service industry, stabilize the price of the nursery service market, formulate laws and policies to standardize and promote the healthy development of nursery-service-related industries as soon as possible, eliminate safety concerns about the nursery service industry, and reduce the economic pressure of fathers on childcare.

### 4.2. Social Level

#### 4.2.1. Promote “Active Aging” and Promote the Healthy Development of the Elderly

We advocate for the elderly to expand their social networks through active social integration and to promote their physical and mental health development. A large number of past studies have shown that whether the elderly take part in cultural and recreational activities, such as singing, playing, illustration, chess, etc., or voluntarily sign up for voluntary activities, such as “helping one another to use smart products” and “guide to sightseeing in red scenic spots”, they can benefit the physical and mental health of the elderly, enhance their physical fitness, and at the same time alleviate their negative emotions, such as depression and anxiety, in the process of caring for their grandchildren. Therefore, while caring for grandchildren, we should also encourage the elderly having time for themselves, actively integrate various activities organized by the community and non-profit organizations, enrich social content outside family care, and adjust their psychological state.

#### 4.2.2. Construct an Incentive Mechanism to Appropriately Encourage the Elderly to Take Care of Each Other

We suggest establishing a space for exclusive activities for the elderly, and at the same time, with the help of online big data, carry out the knowledge forum of “new old age, new care”, and introduce in detail how the elderly can efficiently participate in the care of their grandchildren, deepen the recognition of the value of caring for the elderly from generation to generation, and provide corresponding subsidy support. At the same time, children can not only realize the importance of respecting and loving the elderly in concept but also actively support the elderly through their behavior. Affection is the bond that traditional families rely on to support the elderly. Today, with family sizes shrinking and affection being gradually diluted, the government’s incentive mechanism can repair the rift in family relationships, which makes the intergenerational support of the offspring to their parents not only a form of compensation for parents to provide care for each other, but also a kind of behavior based on maintaining family affection bonds.

### 4.3. Family Level

#### 4.3.1. Respect the Right of the Elderly to Choose Alternate Care and Improve Relevant Legislation

With the improvement of China’s old-age care and medical care for the elderly, views such as “intergenerational care is a sign of the old-age contract” and “grandparents pay more now, and they will feel more at ease when they need their children’s care when they are old” are no longer convincing. The elderly should have the right to choose intergenerational care to be free from bondage and fatigue. At present, there are still gaps in the laws related to the rights and interests of the elderly in China. Acts and people who violate this law are not binding. At the same time, to meet the needs of rural old-age security, legislation should be made in rural old-age security, and legal knowledge should be publicized in rural areas so that children can understand the law and better perform their legal duties.

#### 4.3.2. Carry Forward Traditional Filial Piety Culture and Create a Harmonious Intergenerational Environment

With the sustained development of China’s economy and the continuous improvement of social welfare security, the government has increased the basic old-age insurance for the elderly in urban and rural areas year by year, and the demand for their children’s financial support is decreasing day by day. The elderly are more eager to receive spiritual comfort and life care from their children and grandchildren, such as through accompanying them and having a hot meal with their families. Filial piety is the traditional virtue of the Chinese nation. It is the legal responsibility and moral obligation of children to provide their parents with financial support, daily care, and communication within their abilities. However, the elderly providing intergenerational care for their children cannot simply be used as exchange compensation, so as to truly achieve a two-way interaction between family generations, create a virtuous circle and a more harmonious family intergenerational environment.

## Figures and Tables

**Figure 1 ijerph-20-01441-f001:**
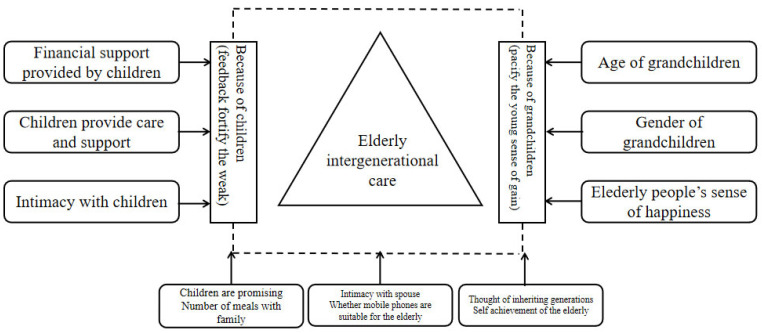
Analysis framework of intergenerational care motivation.

**Figure 2 ijerph-20-01441-f002:**
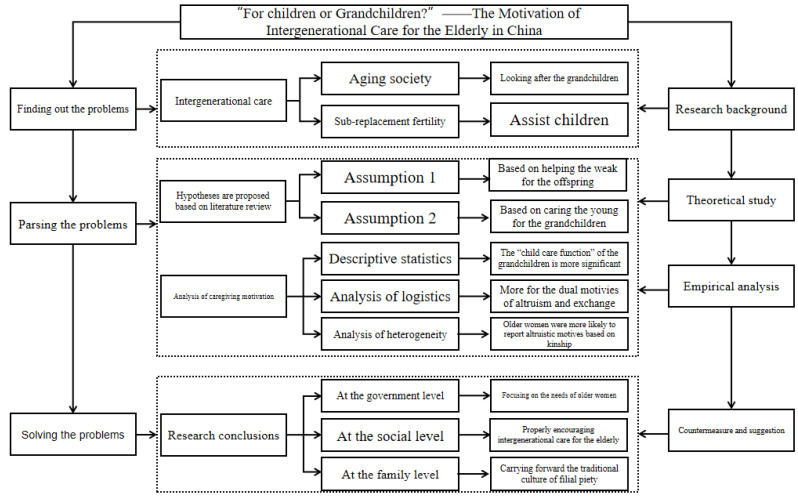
Technology roadmap.

**Table 1 ijerph-20-01441-t001:** Variable selection and overall description.

Variable Category	Variable Name	Assignment Situation and Related Explanation
Explained variable	Day care Evening care	Assign a binary variable: no is 0; yes is 1
Main explanatory variable (because of “zi”)	Provide financial support for children	Binary variable: yes is 1; no is 0
	Provide financial support for children	Binary variable: yes is 1; no is 0
	Intimacy with children	Assign a variable: not close to—1; not too close to—2; generally—3; close to—4; very close to—5
Main explanatory variable (Because of “Sun”)	Sun Dai age	Assign a variable: 0–6 years old—1; 7–13 years old—2; 14–17 years old—3
	Grandchild’s gender	Binary variable: male is 1; female is 0
	Happiness of the elderly	Assign a variable: unimportant—1; not too important—2; generally—3; very important—4
Control variable	Children are promising	Assign a variable: unimportant—1; not too important—2; generally—3; quite important—4; very important—5
	Number of meals a week with family	Continuous variable
	How many boys do you want?	Continuous variable
	The thought of inheriting the family	Assign a variable: unimportant—1; not too important—2; generally—3; quite important—4; very important—5
	Sun Daihu account	Binary variable: town is 1; rural area is 0
	Self-achievement of the elderly	Assign a variable: unimportant—1; not too important—2; generally—3; quite important—4; very important—5
	Gender of the elderly	Binary variable: male is 1; female is 0
	Household registration of the elderly	Binary variable: town is 1; rural area is 0
	Education level of the elderly	Assign a variable: illiterate/semi-illiterate—1; primary school—2; junior high school—3; high school/technical secondary school/technical school/vocational high school—4; college—5; undergraduate degree—6; master’s degree—7; doctor—8
	Be intimate with your spouse	Assign a variable: unimportant—1; not too important—2; generally—3; quite important—4; very important—5
	Do you use a mobile phone?	Binary variable: yes is 1; no is 0

**Table 2 ijerph-20-01441-t002:** Objective fact statistics of table variables.

Variable	Filter Variable Value	Average/Mean Value	Standard Deviation	Minimum Value	Maximum
Day care	1267	0.425	0.494	0	one
Evening care	1267	0.370	0.483	0	one
Provide financial support for children	1267	0.499	0.500	0	one
Provide childcare support	1267	0.381	0.486	0	one
Intimacy with children	1267	4.279	0.697	three	five
Sun Dai age	1267	1.915	0.633	one	three
Grandchild’s gender	1267	0.500	0.500	0	one
Happiness of the elderly	1267	3.157	0.791	one	four
Children are promising	1267	4.640	0.694	one	five
Number of meals a week with family	1267	6.520	1.561	0	seven
How many boys do you want?	1267	1.410	0.703	0	eight
The thought of inheriting the family	1267	4.500	0.861	one	five
Sun Daihu account	1267	0.460	0.498	0	one
Self-achievement of the elderly	1267	3.940	1.091	one	five
Gender of the elderly	1267	0.533	0.499	0	one
Household registration of the elderly	1267	0.450	0.498	0	one
Education level of the elderly	1267	2.133	1.119	one	six
Intimacy with spouse	1267	4.240	1.077	one	five
Do you use a mobile phone?	1267	0.796	0.403	0	one

**Table 3 ijerph-20-01441-t003:** Influence of “child” care: regression results.

Variable	Model 1		Model 2		Model 3	
	Coefficient	Standard Error	Coefficient	Standard Error	Coefficient	Standard Error
Provide financial support for children	0.424 ***	0.121			0.454 ***	0.122
Provide childcare support	−0.468 ***	0.124			−0.493 ***	0.123
Intimacy with children	−0.007	0.087			−0.006	0.089
Sun Dai age			−0.431 ***	0.094	−0.459 ***	0.095
Grandchild’s gender			−0.044	0.117	−0.059	0.118
Happiness of the elderly			−0.032	0.078	−0.019	0.080
Children are promising	−0.227 *	0.096	−0.219 *	0.095	−0.226 *	0.097
Number of meals a week with family	0.094 *	0.040	0.088 *	0.040	0.085 *	0.040
How many boys do you want?	0.055	0.085	0.070	0.085	0.074	0.086
The thought of inheriting the family	−0.014	0.078	−0.007	0.077	−0.020	0.079
Sun Daihu account	−0.212	0.590	−0.344	0.591	−0.327	0.588
Self-achievement of the elderly	0.084	0.062	0.087	0.062	0.088	0.063
Gender of the elderly	−0.005	0.127	0.004	0.127	0.005	0.129
Household registration of the elderly	0.384	0.592	0.439	0.593	0.470	0.590
Education level of the elderly	0.099	0.059	0.060	0.058	0.077	0.059
Intimacy with spouse	0.100	0.063	0.098	0.063	0.110	0.064
Do you use a mobile phone?	0.374 *	0.154	0.342 *	0.154	0.321 *	0.156
(constant)	−1.233 *	0.625	−0.193	0.597	−0.185	0.671
Nagelkerke R Square	0.056		0.054		0.080	

(1) *** *p* < 0.001; (2) * *p* < 0.05.

**Table 4 ijerph-20-01441-t004:** Influencing factors of parental care between different genders.

Variable	Male		Female	
	Coefficient	Standard Error	Coefficient	Standard Error
Provide financial support for children	0.609 ***	0.167	0.280	0.183
Provide childcare support	−0.369 *	0.173	−0.628 ***	0.188
Intimacy with children	0.047	0.121	−0.060	0.133
Sun Dai age	−0.414 *	0.130	−0.570 *	0.145
Grandchild’s gender	0.010	0.162	−0.150	0.176
Happiness of the elderly	−0.196	0.113	0.165	0.119
Children are promising	−0.242	0.134	−0.224	0.144
Number of meals a week with family	0.051	0.053	0.121	0.063
How many boys do you want?	0.027	0.114	0.126	0.133
The thought of inheriting the family	−0.023	0.111	−0.010	0.113
Sun Daihu account	0.053	0.707	−1.042	1.173
Self-achievement of the elderly	0.081	0.085	0.083	0.095
Household registration of the elderly	0.041	0.711	1.251	1.173
Education level of the elderly	0.133	0.078	0.041	0.095
Intimacy with spouse	0.284 **	0.105	0.005	0.083
Do you use a mobile phone?	0.373	0.247	0.248	0.207
(constant)	−0.625	0.948	0.023	0.990
Nagelkerke R Square	0.090		0.097	

(1) *** *p* < 0.001; (2) ** *p* < 0.01; (3) * *p* < 0.05.

**Table 5 ijerph-20-01441-t005:** Influencing factors of parental care in different accounts.

Variable	Cities and Towns		Village	
	Coefficient	Standard Error	Coefficient	Standard Error
Provide financial support for children	0.398 *	0.187	0.469 **	0.167
Provide childcare support	−0.430 *	0.189	−0.618 ***	0.173
Intimacy with children	0.063	0.129	0.056	0.126
Sun Dai age	−0.341 *	0.139	−0.567 **	0.135
Grandchild’s gender	−0.144	0.177	0.017	0.162
Happiness of the elderly	0.070	0.123	−0.092	0.109
Children are promising	−0.265 *	0.133	−0.213	0.145
Number of meals a week with family	0.100	0.060	0.062	0.055
How many boys do you want?	0.103	0.143	0.073	0.109
The thought of inheriting the family	−0.149	0.110	0.090	0.119
Sun Daihu account	0.432	1.259	−0.708	1.716
Self-achievement of the elderly	0.202 *	0.097	0.003	0.084
Household registration of the elderly	−0.046	0.186	0.043	1.184
Education level of the elderly	−0.012	0.085	0.153	0.086
Intimacy with spouse	0.244 *	0.098	0.007	0.087
Do you use a mobile phone?	0.472	0.248	0.179	0.205
(constant)	−0.978	1.591	0.293	0.945
Nagelkerke R Square	0.093		0.096	

(1) *** *p* < 0.001; (2) ** *p* < 0.01; (3) * *p* < 0.05.

**Table 6 ijerph-20-01441-t006:** Effects of table variables on grandparental care at night: regression results.

Variable	1 Model	
	Coefficient	Standard Error
Provide financial support for children	0.459 ***	0.126
Provide childcare support	−0.926 ***	0.132
Intimacy with children	−0.086	0.091
Sun Dai age	−0.314 ***	0.098
Grandchild’s gender	0.139	0.121
Happiness of the elderly	0.000	0.083
Children are promising	−0.157	0.098
Number of meals a week with family	0.034	0.040
How many boys do you want?	0.032	0.089
The thought of inheriting the family	−0.072	0.080
Sun Daihu account	0.522	0.593
Self-achievement of the elderly	0.156 *	0.065
Gender of the elderly	−0.008	0.133
Household registration of the elderly	−0.877	0.596
Education level of the elderly	0.016	0.061
Intimacy with spouse	−0.014	0.065
Do you use a mobile phone?	0.403 *	0.163
(constant)	0.454	0.684
Nagelkerke R Square	0.093	

(1) *** *p* < 0.001; (2) * *p* < 0.05.

## Data Availability

http://www.isss.pku.edu.cn/cfps/download/login.

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
