# Peer review of "For Children or Grandchildren?—The Motivation of Intergenerational Care for the Elderly in China"

_ijerph, 2023, doi:10.3390/ijerph20021441_

Round 1

Reviewer 1 Report

First of all, I am not an expert in quantitative analysis, so I cannot comment on the validity of the analysis methods and results of this study.

I believe that the macroscopic purpose of this research is highly valuable for  this area of study. However, due to the extremely poor literature reviews of this study on inter-generational aid and family relationship, I think the hypotheses and research objectives are too broad and ambiguous to conduct quantitative research and discuss and lead the specific results.
Since this study uses the important data from the region of high interest worldwide, conducting  a more detailed literature review should be conducted.

Author Response

Dear Editor,

     How do you do? Thank you very much for your valuable modification suggestions. I have made the following modifications according to your suggestions and marked them in red.

     Response 1:I put forward the conclusion of this paper more clearly in the abstract.

     Response 2:I have added and improved the literature on intergenerational care from domestic and foreign scholars

     Response 3:I built a technical roadmap for the article to make the context of the article clearer

     Attached is my revised article, please review it again.

Best Wishes.

                                                                                                           Yours,

                                                                                                       Cui Mengmeng  

Reviewer 2 Report

Thank you very much for the opportunity to review this paper, which is of great academic value. In this study, the authors found that offspring are a significant factor in older people's desire for long-term care. One significance of this paper is that it revealed the influence of offspring as well as grandchildren; a second significance is that it showed the mechanism in the context of China, which, for example, reflects a Confucian culture.

I do not think any major changes are necessary, but I would be grateful if you could correct a few small points.

First, the authors should include more quantitative information in the abstract.

Secondly, the process by which the participants in this study were determined is not described. A flow diagram would be helpful to understand this better.

Author Response

Dear Editor,

     How do you do? Thank you very much for your valuable modification suggestions. I have made the following modifications according to your suggestions and marked them in red.

     Response 1:I put forward the conclusion of this paper more clearly in the abstract, and the quantitative methods used in this paper are described in the abstract

     Response 2:I have added and improved the literature on intergenerational care from domestic and foreign scholars

     Response 3:I built a technical roadmap for the article to make the context of the article clearer

     Attached is my revised article, please review it again.

Best Wishes.

                                                                                                           Yours,

                                                                                                       Cui Mengmeng